# Human Papillomavirus Infections and Increased Risk of Incident Osteoporosis: A Nationwide Population-Based Cohort Study

**DOI:** 10.3390/v15041021

**Published:** 2023-04-21

**Authors:** Kevin Sheng-Kai Ma, Ning-Chien Chin, Ting-Yu Tu, Yao-Cheng Wu, Hei-Tung Yip, James Cheng-Chung Wei, Ren-in Chang

**Affiliations:** 1Department of Epidemiology, Harvard T.H. Chan School of Public Health, Boston, MA 02115, USA; 2Center for Global Health, Perelman School of Medicine, University of Pennsylvania, Philadelphia, PA 19104, USA; 3Department of Orthodontics and Dentofacial Orthopedics, Henry M. Goldman School of Dental Medicine, Boston University, Boston, MA 02118, USA; 4Division of Pharmacoepidemiology and Pharmacoeconomics, Department of Medicine, Brigham and Women’s Hospital, Harvard Medical School, Boston, MA 02115, USA; 5Department of Dermatology, Massachusetts General Hospital, Boston, MA 02114, USA; 6Department of Orthopedics, Taichung Veterans General Hospital, Taichung 407, Taiwan; 7Department of Orthopedics, Kaohsiung Veterans General Hospital, Kaohsiung 813, Taiwan; 8School of Medicine, Chung Shan Medical University, Taichung 402, Taiwan; 9Management Office for Health Data, China Medical University Hospital, Taichung 404, Taiwan; 10Institute of Medicine, Chung Shan Medical University, Taichung 402, Taiwan; 11Graduate Institute of Integrated Medicine, China Medical University, Taichung 404, Taiwan; 12Division of Allergy, Immunology and Rheumatology, Department of Internal Medicine, Chung Shan Medical University Hospital, Taichung 402, Taiwan; 13Department of Recreation Sports Management, Tajen University, Pingtung 907, Taiwan; 14Department of Emergency Medicine, Kaohsiung Veterans General Hospital, Kaohsiung 813, Taiwan

**Keywords:** human papillomavirus, osteoporosis, cohort study, viral infections

## Abstract

Patients with viral infections are susceptible to osteoporosis. This cohort study investigated the correlation between human papillomavirus (HPV) infections and the risk of osteoporosis via 12,936 patients with new-onset HPV infections and propensity score-matched non-HPV controls enrolled in Taiwan. The primary endpoint was incident osteoporosis following HPV infections. Cox proportional hazards regression analysis and the Kaplan-Meier method was used to determine the effect of HPV infections on the risk of osteoporosis. Patients with HPV infections presented with a significantly high risk of osteoporosis (adjusted hazard ratio, aHR = 1.32, 95% CI = 1.06–1.65) after adjusting for sex, age, comorbidities and co-medications. Subgroup analysis provided that populations at risk of HPV-associated osteoporosis were females (aHR = 1.33; 95% CI = 1.04–1.71), those aged between 60 and 80 years (aHR = 1.45, 95% CI = 1.01–2.08 for patients aged 60–70; aHR = 1.51; 95% CI = 1.07–2.12 for patients aged 70–80), and patients with long-term use of glucocorticoids (aHR = 2.17; 95% CI = 1.11–4.22). HPV-infected patients who did not receive treatments for HPV infections were at a greater risk (aHR = 1.40; 95% CI = 1.09–1.80) of osteoporosis, while the risk of osteoporosis in those who received treatments for HPV infections did not reach statistical significance (aHR = 1.14; 95% CI = 0.78–1.66). Patients with HPV infections presented with a high risk of subsequent osteoporosis. Treatments for HPV infections attenuated the risk of HPV-associated osteoporosis.

## 1. Introduction

Osteoporosis affects more than 200 million people and causes 8.9 million fractures annually [1], mainly in postmenopausal women [2] and in elderly man [3]. Subsequent hospitalization and surgery not only diminish quality of life but superimpose a growing economic burden on the health care system [4]. Apart from well-established risk factors including endocrine disorders, lifestyle determinants such as low physical activity, sleep disorders [5,6], and the use of glucocorticoids [1,7], the effect of immune system dysregulation on bone turnover rate was revealed to be involved in the regulation of the immune–skeletal interface, for which osteoporosis has been demonstrated to be associated with chronic immune-mediated diseases and inflammatory-related disorders [8]. Abundant cytokines, transcription factors, and cell networks, such as inflammatory cytokines including interleukin (IL)-1, IL-6, IL-17, tumor necrosis factor-alpha (TNF-α) and immunocytes including T cells, B cells, and dendritic cells, participated in the crosstalk of signaling pathways underlying the pathogenesis of severe bone loss and osteoporosis [9].

As a trigger of the immune-mediated mechanisms that could affect bone quality, increasing studies have demonstrated the relationships between low bone mineral density and infections by viruses including human immunodeficiency virus (HIV) [10], hepatitis B virus (HBV) [10,11], hepatitis C virus (HCV) [10], herpes zoster virus (VZV) [12], and severe acute respiratory syndrome coronavirus 2 (SARS-CoV-2) infections [13,14]. For instance, HIV infections were found to elevate the B cell receptor activator of nuclear factor kappa-B ligand (RANKL) and diminish osteo-protegerin, which drives bone resorption [10]. Overall, perturbated by external factors such as acquired immune deficiency syndrome and chronic viral infections, there were trends toward an immunological pattern of immune senescence in osteoporosis, as characterized by the accumulation of activated cells and memory/effector lymphocytes secreting pro-inflammatory cytokines [9], which supported a viral etiology of altered systemic bone turnover rates driven by immune dysfunction.

As some of the most prevalent yet less addressed viral infections in the field of research on osteoporosis, human papillomavirus (HPV) infections are the most commonly diagnosed sexually transmitted disease [15], with more than 600,000 new cases per year around the world [16]. With an HPV-associated etiology, cervical cancer is the second most prevalent cancer among women worldwide and is the most common cancer among women in developing countries [17]. It is well established that HPV infections could be the primary cause of cervical cancers [18] and are also associated with the risk of head and neck cancer, oral cavity cancer, lung cancer, esophageal cancer, breast cancer, and anogenital lesions [19]. Effective approaches for the prevention of HPV-associated cancers include 2vHPV and 4vHPV vaccinations, which have been shown to effectively prevent high-grade intraepithelial lesions in the female genital area [20], the mechanism of which was related to the blocked immune evasion strategies of HPV infections, including alterations in gene expression, protein function, and antigen processing, which would have led to carcinogenesis in unvaccinated individuals [21].

HPV infections and other viral infections share several immune dysfunction pathogeneses that could contribute to the development of osteoporosis, which include a strongly elevated RANKL expression in the progression of HPV-associated cervical neoplasia [22]. Bone resorption and osteoclast formation are driven by RANKL/RANK signaling [23]. The binding between RANKL and RANK activates NF-κB and c-Fos, leading to osteo-clastogenesis. Moreover, it was found that excessive RANKL production by HPV-infected cells could both aggravate tumor burden and exacerbate osteoclast-mediated bone destruction [24]. As such, it was investigated in the present nationwide population-based cohort study as to whether HPV infections could be associated with subsequent risk of osteoporosis or osteoporotic fracture.

## 2. Materials and Methods

### 2.1. Data Source

The present population-based cohort study was performed using data from the National Health Insurance Research Database (NHIRD). The NHIRD contains registration files and original claim data, such as hospitalization records, diagnostic codes, records of outpatient visits, medication history, and personal information for over 99% of Taiwan’s population. All diagnostic records were validated by the Bureau of National Health Insurance (NHI) to ensure the accuracy of these data [25]. In this study, data were retrieved from the Longitudinal Health Insurance Research Database (LHIRD), a subset of NHIRD, which was composed of claims data of one million people randomly sampled from the study period between 1997 and 2013. The Institutional Review Board of Taichung Veterans General Hospital approved this study.

### 2.2. Study Population

In the present study, patients who had been clinically diagnosed with HPV infections between 2000 and 2013 were identified using International Classification of Disease Clinical Modification codes (ICD-9-CM code 079.4, 078.1, 078.10–078.12, 078.19, 759.05, 795.09, 795.15, 795.19, 796.75 and 796.79) [26,27]. Only patients with diagnosis of HPV from at least one inpatient admission or three outpatient visits were selected. The index date was defined as the date of HPV infection diagnosis. Individuals with osteoporosis before the index date or within 1 year after the index date were excluded. Patients aged under 50 years old or beyond 100 years old, or patients with missing information on gender were excluded. The control group was selected from LHIRD, propensity score-matched in a 1:4 ratio by age, sex, index year, co-morbidities and co-medications, using the same protocol as described in previous studies [28,29,30,31]. In the subgroup analysis, patients with HPV infections who had received treatment procedures within three months after the index date were categorized as “with treatment” group, and otherwise as “without treatment” group. Such treatment procedures included: (1) electrocauterization for condyloma, (2) condyloma, excision, and electrocauterization, (3) CO_2_ laser operation, (4) chemosurgery for condyloma, (5) simple or complicated electro cauterization, (6) liquid nitrogen cryosurgery, (7) simple or complicated cryotherapy that involved CO_2_ freezing and liquid nitrogen. All participants were tracked until presence of osteoporosis, missing, death, or the end of the study, December 2013. A total of 12,936 subjects were included in the HPV group and 51,744 subjects were included in the non-HPV control group.

### 2.3. Primary Endpoint and Covariates

All subjects in both groups were tracked from the index date to the first osteoporosis event. The primary endpoint of this study was the occurrence of new-onset osteoporosis, which was diagnosed based on clinical history or spinal or hip bone mineral density (BMD) by dual-energy X-ray absorptiometry (DEXA) evaluation and the use of one of the following medications: denosumab, alendronate, risedronate, ibandronate, zoledronate, raloxifene, bazedoxifene, teriparatide, strontium ranelate [32]. Patients diagnosed with osteoporosis or the use of aforementioned medications before the index date or within one year after the index date were excluded. The use of osteoporosis medications as alternative representatives of osteoporosis patients, in addition to BMD or DEXA assessment, was aimed to intensify the diagnostic validity. To calculate the co-payment exemption of these medications, the NHI bureau used an auditing mechanism to minimize diagnostic uncertainty and misclassification of osteoporosis. To eliminate potential bias, factors including the demographic variables and relevant co-morbidities including hypertension, diabetes mellitus, hyperlipidemia, rheumatoid arthritis, chronic obstructive pulmonary disease (COPD), alcohol-related illness, chronic kidney disease (CKD), inflammatory bowel disease (IBD), HBV, HCV, cirrhosis, celiac disease, syphilis, HIV, chlamydia, gonococcus, hyperthyroidism, hyperparathyroidism, vitamin D deficiency, premature menopause, male hypogonadism, adrenal cortical steroids, smoking, and bone mineralization affecting [33] co-medications including long-term glucocorticoid use [34], phenobarbital, phenytoin, carbamazepine, heparin, warfarin, cyclosporine, tricyclic antidepressants (TCAs) or selective serotonin reuptake inhibitors (SSRIs), proton pump inhibitors, furosemide, thiazide, statin, and beta-blockers. Baseline characteristics within two years before the index date were retrieved and adjusted in the analyses.

### 2.4. Statistical Analysis

The chi-square test was used to compare the distribution of age, sex, index date, and baseline co-morbidities between the two groups, including those with versus without a previous history of HPV infections. The mean age of onset was compared using Student’s t-test. The incidence rate of osteoporosis was estimated by dividing the amount of osteoporosis by follow-up person-years for HPV-infected cases and non-HPV controls. Multivariable Cox proportional hazards regression models were used to estimate the crude HRs (cHRs) and adjusted HRs (aHRs) for osteoporosis in patients with HPV infections, as compared to non-HPV controls. The covariates considered in the multivariable regression models included sex, age, index date, comorbidities, and co-medications. A multivariable Cox regression model adjusted for the covariates was used to reveal the effect of sex, age, and follow-up time on the incidence of osteoporosis. The HRs adjusted for covariates were calculated each subgroup, as stratified by sex, age, comorbidities, and co-medications. The cumulative incidence curve was obtained from the Kaplan-Meier method and examined by the log-rank test. All data analyses were performed using SAS (version 9.4; SAS Institute, Inc., Carey, NC, USA). The statistical significance level was set at *p*-value < 0.05 in the two-tailed test.

## 3. Results

### 3.1. Basline Characteristics of Study Populations

In this retrospective cohort study, 12,936 HPV-infected patients and 51,744 propensity score-matched non-HPV controls were included. Baseline characteristics are shown in Table 1. The mean ages of the patients in the HPV and non-HPV groups were 63.26 and 63.13 years, respectively.

### 3.2. Incidence of Osteoporosis in Patients with HPV Infections

The cumulative incidence of osteoporosis in patients with HPV was significantly higher than that in non-HPV controls (log-rank test *p*-value = 0.01) (Figure 1). The Cox proportional hazards regression models revealed that patients with HPV infections presented with significantly greater risk of osteoporosis (aHR = 1.32, 95% CI = 1.06–1.65) (Table 2). Consistent with previous studies [1,35,36], the risk of osteoporosis was lower in men (aHR = 0.23; 95% CI = 0.18–0.3), and in populations of high socioeconomic status [37] (aHR = 0.30; 95% CI = 0.14–0.64); while the risk of osteoporosis was higher in frequent out-patient department (OPD) visitors (aHR = 1.01; 95% CI = 1.01–1.01) and in patients with comorbidities including COPD (aHR = 1.35; 95% CI = 1.08–1.68), vitamin D deficiency (aHR = 18.3; 95% CI = 2.56–131.09), long term glucocorticoid use (aHR = 2.05; 95% CI = 1.44–2.92), and TCAs/SSRIs use (aHR = 1.60; 95% CI = 1.05–2.43). The risk of osteoporosis was also higher in the older age groups [38], in which patients aged between 60–70 (aHR = 4.29; 95% CI = 3.10–5.93), aged between 70–80 (aHR = 7.32; 95% CI = 5.23–10.26) and aged beyond 80 (aHR = 9.63; 95% CI = 6.22–14.88) were associated with a significantly high risk of osteoporosis (Table 2).

### 3.3. Factors Associated with HPV-Associated Osteoporosis

Findings in the subgroup analysis of Cox proportional hazards regression models provided that female patients with HPV infections (aHR = 1.33; 95% CI = 1.04–1.71), patients aged 60–70 (aHR = 1.45; 95% CI = 1.01–2.08) or aged 70–80 (aHR = 1.51; 95% CI = 1.07–2.12) with HPV infections, patients of the lowest (aHR = 1.32; 95% CI = 1.01–1.73) or the highest socioeconomic status (aHR = 8.22; 95% CI = 1.59–42.38) with HPV infections, and HPV-infected patients with long-term use of glucocorticoids (aHR = 2.17; 95% CI = 1.11–4.22) were predisposed to significantly great risk of HPV-associated osteoporosis (Table 3). Notably, HPV-infected patients who did not receive treatments for HPV infections had a significantly great risk of osteoporosis (aHR = 1.40; 95% CI = 1.09–1.80), compared to non-HPV controls; on the other hand, the risk of osteoporosis in HPV-infected patients who received treatments for HPV infections did not reach statistical significance (aHR = 1.14; 95% CI = 0.78–1.66) (Table 4). Overall, treatments for HPV infections attenuated the risk of osteoporosis in patients with HPV infections.

## 4. Discussion

In this 13-year nationwide population-based retrospective cohort study, there was a 32% increased risk of osteoporosis in patients with HPV infections, which was validated after adjusting for demographic variables, comorbidities, and co-medications. Patients who were women, aged between 60 and 80, of low and of high socioeconomic status, with long-term use of glucocorticoid, were susceptible to HPV-associated osteoporosis. Treatments for HPV infections lowered the risk of HPV-associated osteoporosis in patients with HPV infections.

Studies have shown that virus infections, such as HIV, HBV, HCV, and herpes zoster infections, were independently associated with a higher risk of osteoporosis [10,11]. Collectively, evidence from these studies provided the association between viral infections and the risk of osteoporosis. For instance, it was demonstrated that patients with herpes zoster infections presented with a 4.55-fold greater risk of osteoporosis [12], as associated with significantly high levels of interleukin (IL)-1b, IL-6, IL-8, IL-10, and tumor necrosis factor-alpha (TNF-α) [39]. Among those inflammatory biomarkers, IL-6 was suggested as a potent stimulator of osteoclast-induced bone resorption and thus was central to the pathogenesis of bone loss in the context of chronic inflammation [40]. Likewise, HIV-infected individuals were reported to have lower BMD compared to non-infected controls [41], as supported by a meta-analysis that observed a significant increase in the risk of fractures in HIV-infected individuals with an incidence rate ratio of 1.58 (95% CI = 1.25–2.00) [41]. It was further demonstrated that a B-cell RANKL/osteo-protegerin-driven pathogenesis contributed to the compromised total hip and femoral neck BMD in ART-naïve HIV-infected patients, indicating that B-cell dysregulation promoted HIV-induced bone loss through an imbalance in the RANKL/osteo-protegerin ratio [42]. In the case of HBV infections, patients with HBV infections were shown to present with a significant great risk of osteoporosis [11], which was in accordance with observations that the seropositivity of the surface antigen for hepatitis B in adult men was significantly associated with lower BMD [43]. Chronic HBV infections can induce the production of inflammatory cytokines, such as TNF-α, IL-1, and IL-6, which increases RANKL that stimulates osteo-clastogenesis and bone resorption [44]. Moreover, TNF-α can inhibit osteoblast differentiation and promote osteoblast apoptosis [45], for which HBV infection-associated osteoporosis was proposed to be driven by inflammatory pathways that contributed to decreased bone formation, increased bone resorption, and a subsequently decreased systemic BMD [11].

The exact mechanism of how HPV infections undermined bone loss or osteoporosis has not been studied. However, highly expressed RANKL has been observed during the progression of HPV infection-associated cervical cancer, which was secreted by HPV-infected cells [22]. The excessive production of RANKL released by tumor cells was reported to trigger osteoclast-mediated bone destruction and to increase tumor burden [24]. Subsequently, binding of RANKL to RANK receptors on osteoclasts was found to activate signals for bone resorption [22]. On the other hand, high levels of inflammatory mediators such as TNF-α in patients with HPV infections [46] could place those individuals in an inflammatory microenvironment that may intensify osteoclastic resorption [47] by promoting RANKL production [48], transducing RANKL-induced signal pathways, and amplifying osteo-clastogenesis [47,49]. Specifically, TNF-α, as triggered by infections, promotes osteoblasts apoptosis and reduces osteo-blastogenesis by stimulating DKK-1 and Sost expression [9]. Moreover, TNF-α could suppress osteoblast differentiation by inhibiting Smad signaling through an NF-κB-mediated process [50]. Collectively, osteo-clastogenesis in response to high concentrations of RANKL and TNF-α may explain bone resorption and osteoporosis in patients with HPV infections.

It was found in the present study that treatments for HPV infections attenuated the risk of HPV infections. This finding was in accordance with previous studies on benign lesions of anogenital warts [51], in which medical and surgical therapies were able to alleviate symptoms [52]. For instance, the primary clearance rate of lesions was estimated to be 44–87% for cryotherapy, 89–100% for scissor excision, 94–100% for electrocautery, and close to 100% for laser-assisted surgical treatments [53]. Moreover, as most HPV infections with genital warts could be eradicated within two years in immune-competent patients, early treatments of warts has been shown to exert higher clearance rates and lower incidence of malignancies [54], which may explain the observed beneficial effect of treatments for HPV infections on the reduced the risk of osteoporosis in the present study. However, as treatments may not eradicate all HPV-infected cells, long-term follow-ups for patients with HPV infections would still be necessary. In particular, it was demonstrated that there was a recurrence rate of 20–30% after therapies on HPV-associated wart lesions, which could increase during follow-ups [53]. The recurrence rate of wart lesions after treatments for HPV infections was 12–42% at 1 to 3 months and 59% at 12 months following cryotherapy, 9–29% following scissor excision, 22% following electrocautery, and 17–19% at 3 months and 66% at 12 months following laser surgery [53]. All in all, early treatments for HPV infections and long-term follow-ups were recommended. Clinical implications also included screening [55] in patients with HPV infections and patient education [56,57,58,59,60,61].

The major strength of the present study was the use of longitudinal data of large sample size and a long follow-up duration that was able to provide the temporal association between HPV infections and the risk of new-onset osteoporosis, and the effect of treatments for HPV infections on the attenuated risk of HPV-associated osteoporosis. Findings in the present cohort study were representative of the general population, and potential measurable confounders were balanced through propensity score matching [58,59,60] for demographics, comorbidities, and co-medications. In addition, both the diagnoses of HPV infections and osteoporosis were adjudicated by physicians, which ascertained the accuracy of the diagnoses. The outcome measurement of osteoporosis was further validated via requiring the use of medications for osteoporosis as part of the criteria. In addition, subgroup analyses were used to elucidate effect measure modifications. That said, several limitations exist in this study. First, there could be underestimated cases of HPV infections if there were no clinically recognized lesions, which could have excluded patients with self-resolving or asymptomatic infections from the study population. Second, most of the Taiwanese population are East Asians, so findings of the present study may not be generalizable to other populations. Third, as information on serotypes of HPV, history of HPV vaccinations, and lifestyle determinants such as physical activity were not available in the dataset, further studies are warranted to address whether these factors could alter the risk of HPV-associated osteoporosis. Finally, the differentiation between HPV positivity alone and pre-cancerous lesions are not clarified. A stratified analysis based on the degree of lesion needs further execution.

## 5. Conclusions

In conclusion, patients with HPV infections were associated with a significantly greater risk of subsequent osteoporosis, especially in female patients, those aged between 60 and 80 years, and individuals with long-term use of glucocorticoids. Treatments for HPV attenuated the risk of HPV-associated osteoporosis. Prospective studies on the association between HPV genotypes and osteoporosis and studies of mechanistic approaches may shed light on HPV-associated bone loss and osteoporosis.

## Figures and Tables

**Figure 1 viruses-15-01021-f001:**
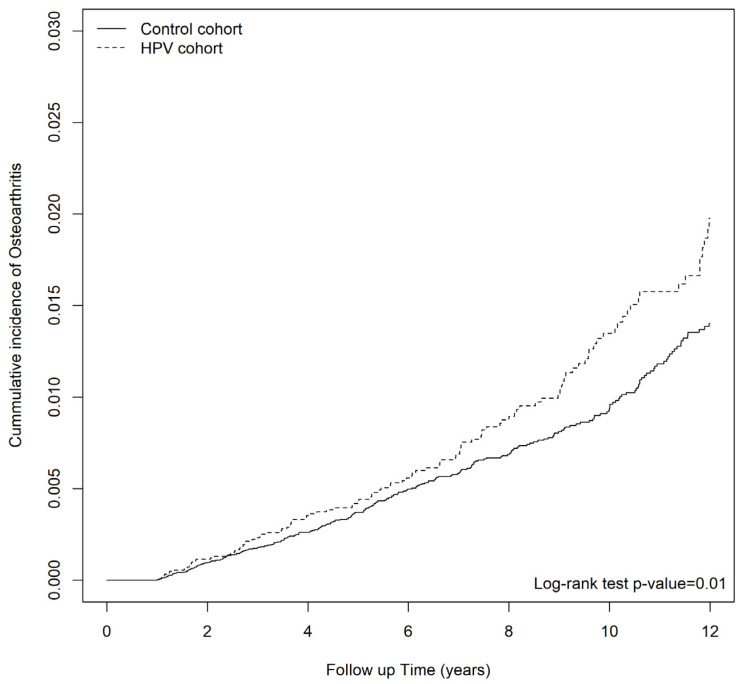
Kaplan-Meier curves for the cumulative incidence of new-onset osteoporosis in individuals with or without human papillomavirus infections.

**Table 1 viruses-15-01021-t001:** Baseline characteristics in HPV-infected patients and non-HPV controls.

	HPV				
	No (*N* = 51,744)		Yes (*N* = 12,936)		
Variables	*n*	%	*n*	%	SMD
Gender					0.031
Female	25,596	49.47	6198	47.91	
Male	26,148	50.53	6738	52.09	
Age, year					
50–60	24,335	47.03	5958	46.06	0.019
60–70	14,546	28.11	3496	27.03	0.024
70–80	9547	18.45	2661	20.57	0.054
>80	3316	6.41	821	6.35	0.003
mean, (SD)	63.13	(9.81)	63.26	(9.87)	0.013
Socioeconomic status (Monthly salaries in New Taiwan Dollar)					
<20,000	25,843	49.94	6835	52.84	0.058
20,001–40,000	17,163	33.17	3754	29.02	0.090
>40,000	8738	16.89	2347	18.14	0.033
Out-patient visit frequency mean, (SD)	30.8	(26.3)	32.2	(23.8)	0.055
Comorbidities					
Hypertension	30,535	59.01	7460	57.67	0.027
Diabetes	17,663	34.14	4330	33.47	0.014
Hyperlipidemia	22,185	42.87	5406	41.79	0.022
COPD	15,937	30.80	3947	30.51	0.006
IBD	2871	5.55	713	5.51	0.002
HBV	3050	5.89	748	5.78	0.005
HCV	1370	2.65	346	2.67	0.002
Cirrhosis	25,165	48.63	6050	46.77	0.037
Celiac disease	5	0.01	1	0.01	0.002
RA	3654	7.06	916	7.08	0.001
CKD	2595	5.02	639	4.94	0.003
Syphilis	261	0.50	65	0.50	<0.001
HIV	60	0.12	17	0.13	0.004
Chlamydia	2	0.004	0	-	0.009
Gonococcal	172	0.33	38	0.29	0.007
Hyperthyroidism	2388	4.62	596	4.61	<0.001
Hyperparathyroidism	73	0.14	20	0.15	0.004
Vitamin D deficiency	17	0.03	3	0.02	0.006
Premature menopause	1	0.002	1	0.01	0.008
Male hypogonadism	23	0.04	9	0.07	0.011
Adrenal cortical steroids	1	0.002	0	-	0.006
Smoking	699	1.35	182	1.41	0.005
Alcohol	2088	4.04	521	4.03	<0.001
Co-medications					
Long-term use of glucocorticoids	2011	3.89	526	4.07	0.009
Phenobarbital, phenytoin, or carbamazepine	368	0.71	99	0.77	0.006
Heparin or warfarin	418	0.81	101	0.78	0.003
Cyclosporine	40	0.08	9	0.07	0.003
TCAs or SSRIs	1662	3.21	435	3.36	0.008
PPIs	1676	3.24	444	3.43	0.011
Furosemide	1431	2.77	348	2.69	0.005
Thiazide	1816	3.51	457	3.53	0.001
Statin	5327	10.29	1338	10.34	0.002
Beta blockers	9682	18.71	2422	18.72	<0.001

SMD, Standard mean difference; COPD, Chronic obstructive pulmonary disease; IBD, Inflammatory bowel disease; HBV, Hepatitis B virus; HCV, Hepatitis C virus; HIV, Human Immunodeficiency Virus; RA, Rheumatoid arthritis; CKD, Chronic kidney disease; TCAs, Tricyclic antidepressants; SSRIs, Selective serotonin receptor inhibitors; PPIs, proton pump inhibitors.

**Table 2 viruses-15-01021-t002:** Factors associated with a diagnosis of osteoporosis in the Cox proportional hazard models.

	Osteoporosis						
Variable	*n*	PY	IR	cHR	(95% CI)	aHR	(95% CI)
HPV infections							
non-HPV	277	302,947	0.91	1.00		1.00	
HPV	116	91,893	1.26	1.33	(1.07, 1.65) *	1.32	(1.06, 1.65) *
Gender							
Female	315	197,533	1.59	1.00		1.00	
Male	78	197,307	0.40	0.25	(0.19, 0.32) ***	0.23	(0.18, 0.3) ***
Age, year							
50–60	51	196,324	0.26	1.00		1.00	
60–70	150	114,244	1.31	5.04	(3.67, 6.92) ***	4.29	(3.10, 5.93) ***
70–80	151	68,370	2.21	9.05	(6.59, 12.4) ***	7.32	(5.23, 10.26) ***
>80	41	15,902	2.58	12.7	(8.42, 19.3) ***	9.63	(6.22, 14.88) ***
Socioeconomic status (Monthly salaries in New Taiwan Dollar)							
<20,000	249	195,413	1.27	1.00		1.00	
20,001–40,000	137	132,988	1.03	0.79	(0.64, 0.98) *	1.06	(0.86, 1.32)
>40,000	7	66,439	0.11	0.08	(0.04, 0.18) ***	0.30	(0.14, 0.64) **
Out-patient visit frequency				1.02	(1.01, 1.02)	1.01	(1.01, 1.01) ***
Comorbidities							
Hypertension							
No	126	179,675	7.01	1.00		1.00	
Yes	267	215,165	12.41	1.89	(1.53, 2.34) ***	0.88	(0.69, 1.12)
Diabetes							
No	237	275,713	8.60	1.00		1.00	
Yes	156	119,127	13.10	1.63	(1.33, 2.00) ***	1.00	(0.8, 1.24)
Hyperlipidemia							
No	234	256,496	9.12	1.00	(reference)	1.00	(reference)
Yes	159	138,343	11.49	1.47	(1.20, 1.81) ***	0.98	(0.77, 1.24)
COPD							
No	241	294,227	8.19	1.00	(reference)	1.00	(reference)
Yes	152	100,612	15.11	2.10	(1.71, 2.58) ***	1.35	(1.08, 1.68) **
IBD							
No	380	377,908	10.06	1.00	(reference)		
Yes	13	16,931	7.68	0.86	(0.50, 1.50)		
HBV							
No	384	378,125	10.16	1.00	(reference)		
Yes	9	16,714	5.38	0.62	(0.32, 1.2)		
HCV							
No	385	387,298	9.94	1.00	(reference)		
Yes	8	7541	10.61	1.25	(0.62, 2.52)		
Cirrhosis							
No	213	219,518	9.70	1.00	(reference)		
Yes	180	175,322	10.27	1.13	(0.92, 1.37)		
Celiac disease							
No	393	394,806	9.95	1.00	(reference)		
Yes	0	33	0.00	0.00	(0, Inf)		
RA							
No	360	371,950	9.68	1.00	(reference)	1.00	(reference)
Yes	33	22,889	14.42	1.66	(1.16, 2.37) **	0.99	(0.69, 1.43)
CKD							
No	369	381,138	9.68	1.00	(reference)	1.00	(reference)
Yes	24	13,701	17.52	2.11	(1.39, 3.19) ***	1.22	(0.8, 1.88)
Syphilis							
No	392	393,410	9.96	1.00	(reference)		
Yes	1	1430	7.00	0.80	(0.11, 5.68)		
HIV							
No	393	394,460	9.96	1.00	(reference)		
Yes	0	379	0.00	0.00	(0, Inf)		
Chlamydia							
No	393	394,834	9.95	1.00	(reference)		
Yes	0	5	0.00	0.00	(0, Inf)		
Gonococcal							
No	393	393,856	9.98	1.00	(reference)		
Yes	0	983	0.00	0.00	(0, Inf)		
Hyperthyroidism							
No	376	380,661	9.88	1.00	(reference)		
Yes	17	14,178	11.99	1.37	(0.84, 2.23)		
Hyperparathyroidism							
No	392	394,411	9.94	1.00	(reference)		
Yes	1	429	23.33	2.65	(0.37, 18.87)		
Vitamin D deficiency							
No	392	394,734	9.93	1.00	(reference)	1.00	(reference)
Yes	1	105	95.23	10.4	(1.45, 73.69) *	18.3	(2.56, 131.09) **
Premature menopause							
No	393	394,822	9.95	1.00	(reference)		
Yes	0	18	0.00	0.00	(0, Inf)		
Male hypogonadism							
No	393	394,663	9.96	1.00	(reference)		
Yes	0	176	0.00	0.00	(0, Inf)		
Adrenal cortical steroids							
No	393	394,839	9.95	1.00	(reference)		
Yes	0	0	0.00	NA	(NA, NA)NA		
Smoking							
No	391	391,790	9.98	1.00	(reference)		
Yes	2	3049	6.56	0.88	(0.22, 3.56)		
Alcohol							
No	389	384,044	10.13	1.00	(reference)		
Yes	4	10,795	3.71	0.44	(0.16, 1.17)		
Co-medications							
Long-term use of glucocorticoids							
No	355	382,271	9.29	1.00	(reference)	1.00	(reference)
Yes	38	12,568	30.24	3.47	(2.48, 4.85) ***	2.05	(1.44, 2.92) ***
Phenobarbital, phenytoin, or carbamazepine							
No	388	392,217	9.89	1.00	(reference)		
Yes	5	2622	19.07	1.98	(0.82, 4.79)		
Heparin or warfarin							
No	390	392,544	9.94	1.00	(reference)		
Yes	3	2295	13.07	1.51	(0.49, 4.71)		
Cyclosporine							
No	393	394,557	9.96	1.00	(reference)		
Yes	0	282	0.00	0.00	(0, Inf)		
TCAs or SSRIs							
No	368	384,588	9.57	1.00	(reference)	1.00	(reference)
Yes	25	10,251	24.39	2.79	(1.86, 4.19) ***	1.60	(1.05, 2.43) *
PPIs							
No	383	386,939	9.90	1.00	(reference)		
Yes	10	7900	12.66	1.64	(0.87, 3.08)		
Furosemide							
No	376	387,563	9.70	1.00	(reference)	1.00	(reference)
Yes	17	7276	23.37	2.84	(1.74, 4.62) ***	1.14	(0.69, 1.91)
Thiazide							
No	369	382,117	9.66	1.00	(reference)	1.00	(reference)
Yes	24	12,722	18.86	2.05	(1.36, 3.10) ***	1.13	(0.74, 1.74)
Statin							
No	349	365,283	9.55	1.00	(reference)	1.00	(reference)
Yes	44	29,556	14.89	1.86	(1.35, 2.55) ***	1.23	(0.87, 1.73)
Beta blockers							
No	307	326,634	9.40	1.00	(reference)	1.00	(reference)
Yes	86	68,205	12.61	1.40	(1.10, 1.78) **	0.85	(0.65, 1.10)

PY, person-year; IR, Incidence rate; cHR, Crude Hazard ratio; aHR, Adjust Hazard ratio; OPD, Out-patient department; COPD, Chronic obstructive pulmonary disease; IBD, Inflammatory bowel disease; HBV, Hepatitis B virus; HCV, Hepatitis C virus; HIV, Human Immunodeficiency Virus; RA, Rheumatoid arthritis; CKD, Chronic kidney disease; TCAs, Tricyclic antidepressants; SSRIs, Selective serotonin receptor inhibitors; PPIs, proton pump inhibitors; *, *p* < 0.05; **, *p* < 0.01; ***, *p* < 0.001.

**Table 3 viruses-15-01021-t003:** Subpopulations at risk of HPV-associated osteoporosis in patients with HPV infections.

	non-HPV			HPV						
Variable	N	PY	IR	*n*	PY	IR	cHR	(95% CI)	aHR	(95% CI)
Gender										
Female	227	154,018	14.7	88	43,514	20.2	1.33	(1.04,1.70) *	1.33	(1.04,1.71) *
Male	50	148,928	3.4	28	48,378	5.8	1.65	(1.04,2.62) *	1.29	(0.80,2.08)
Age, year										
50–60	41	153,573	2.7	10	42,751	2.3	0.87	(0.43,1.73)	0.88	(0.44,1.76)
60–70	107	88,263	12.1	43	25,980	16.6	1.28	(0.90,1.83)	1.45	(1.01,2.08) *
70–80	97	49,295	19.7	54	19,075	28.3	1.33	(0.95,1.86)	1.51	(1.07,2.12) *
>80	32	11,816	27.1	9	4086	22.0	0.76	(0.36,1.59)	0.95	(0.45,2.01)
Socioeconomic status (Monthly salaries in New Taiwan Dollar)										
<20,000	167	146,307	11.4	82	49,106	16.7	1.39	(1.06,1.81) *	1.32	(1.01,1.73) *
20,001–40,000	108	106,808	10.1	29	26,179	11.1	1.07	(0.71,1.62)	1.22	(0.81,1.85)
>40,000	2	49,832	0.4	5	16,607	3.0	7.47	(1.45,38.56) *	8.22	(1.59,42.38) *
Comorbidities										
Hypertension										
No	86	139,008	6.2	40	40,667	9.8	1.56	(1.07,2.26) *	1.50	(1.02,2.20) *
Yes	191	163,939	11.7	76	51,226	14.8	1.21	(0.93,1.58)	1.27	(0.97,1.67)
Diabetes										
No	163	212,619	7.7	74	63,094	11.7	1.48	(1.13,1.95) **	1.42	(1.07,1.88) *
Yes	114	90,328	12.6	42	28,798	14.6	1.09	(0.77,1.56)	1.22	(0.85,1.74)
Hyperlipidemia										
No	168	197,826	8.5	66	58,670	11.2	1.28	(0.96,1.71)	1.29	(0.97,1.73)
Yes	109	105,120	10.4	50	33,223	15.0	1.36	(0.97,1.90)	1.35	(0.96,1.89)
COPD										
No	171	227,344	7.5	70	66,883	10.5	1.35	(1.02,1.78) *	1.41	(1.06,1.87) *
Yes	106	75,603	14.0	46	25,009	18.4	1.22	(0.86,1.73)	1.24	(0.87,1.77)
IBD										
No	267	29,0246	9.2	113	87,662	12.9	1.35	(1.08,1.68) **	1.36	(1.09,1.70) **
Yes	10	12,700	7.9	3	4231	7.1	0.89	(0.24,3.24)	1.38	(0.36,5.27)
HBV										
No	271	290,425	9.3	113	87,700	12.9	1.33	(1.07,1.65) *	1.33	(1.07,1.67) *
Yes	6	12,521	4.8	3	4193	7.2	1.33	(0.33,5.35)	1.55	(0.38,6.37)
HCV										
No	272	297,353	9.1	113	89,945	12.6	1.32	(1.06,1.64) *	1.33	(1.06,1.66) *
Yes	5	5594	8.9	3	1948	15.4	1.76	(0.42,7.43)	2.46	(0.41,14.64)
Cirrhosis										
No	143	168,617	8.5	70	50,900	13.8	1.58	(1.19,2.11) **	1.58	(1.18,2.12) **
Yes	134	134,329	10.0	46	40,992	11.2	1.06	(0.76,1.48)	1.06	(0.76,1.50)
RA										
No	250	285,616	8.8	110	86,334	12.7	1.40	(1.12,1.75) **	1.40	(1.12,1.76) **
Yes	27	17,330	15.6	6	5558	10.8	0.64	(0.26,1.54)	0.73	(0.29,1.81)
CKD										
No	259	292,591	8.9	110	88,547	12.4	1.35	(1.08,1.68) **	1.34	(1.07,1.68) *
Yes	18	10,355	17.4	6	3346	17.9	1.00	(0.4,2.53)	1.50	(0.56,4.00)
Hyperthyroidism										
No	263	292,224	9.0	113	88,437	12.8	1.36	(1.09,1.70) **	1.37	(1.10,1.72) **
Yes	14	10,723	13.1	3	3455	8.7	0.64	(0.19,2.25)	0.65	(0.18,2.34)
Alcohol										
No	275	294,914	9.3	114	89,130	12.8	1.32	(1.06,1.64) *	1.32	(1.06,1.65) *
Yes	2	8033	2.5	2	2762	7.2	2.77	(0.39,19.84)	0.77	(0.10,5.65)
Co-medications										
Long-term use of glucocorticoids										
No	254	293,679	8.6	101	88,592	11.4	1.26	(1.00,1.59) *	1.27	(1.00,1.6) *
Yes	23	9268	24.8	15	3300	45.4	1.75	(0.91,3.35)	2.17	(1.11,4.22) *
Phenobarbital, phenytoin, or carbamazepine										
No	273	301,023	9.1	115	91,194	12.6	1.34	(1.07,1.66) **	1.35	(1.08,1.68) **
Yes	4	1924	20.8	1	699	14.3	0.63	(0.07,5.7)	0.00	(0,Inf)
Heparin or warfarin										
No	274	301,212	9.1	116	91,332	12.7	1.34	(1.08,1.66) **	1.35	(1.08,1.68) **
Yes	3	1734	17.3	0	561	0.0	0.00	(0,Inf)	0.00	(0,Inf)
TCAs or SSRIs										
No	259	295,564	8.8	109	89,025	12.2	1.34	(1.07,1.68) **	1.35	(1.08,1.70) **
Yes	18	7383	24.4	7	2868	24.4	0.93	(0.39,2.23)	0.99	(0.39,2.47)
PPIs										
No	270	297,150	9.1	113	89,789	12.6	1.33	(1.07,1.66) *	1.34	(1.07,1.67) *
Yes	7	5797	12.1	3	2104	14.3	0.96	(0.25,3.73)	2.08	(0.41,10.71)
Furosemide										
No	266	297,458	8.9	110	90,105	12.2	1.31	(1.05,1.64) *	1.31	(1.04,1.64) *
Yes	11	5488	20.0	6	1787	33.6	1.60	(0.59,4.32)	2.02	(0.70,5.78)
Thiazide										
No	262	293,245	8.9	107	88,872	12.0	1.29	(1.03,1.62) *	1.30	(1.03,1.63) *
Yes	15	9702	15.5	9	3020	29.8	1.87	(0.82,4.27)	2.14	(0.89,5.14)
Statin										
No	246	280,748	8.8	103	84,535	12.2	1.34	(1.06,1.68) *	1.36	(1.07,1.71) *
Yes	31	22,199	14.0	13	7357	17.7	1.18	(0.62,2.26)	1.10	(0.56,2.15)
Beta blockers										
No	210	251,618	8.3	97	75,016	12.9	1.50	(1.18,1.91) **	1.49	(1.17,1.91) **
Yes	67	51,329	13.1	19	16,876	11.3	0.81	(0.49,1.35)	0.91	(0.54,1.52)

PY, person-year; IR, Incidence rate; cHR, Crude Hazard ratio; aHR, Adjust Hazard ratio; OPD, Out-patient department; COPD, Chronic obstructive pulmonary disease; IBD, Inflammatory bowel disease; HBV, Hepatitis B virus; HCV, Hepatitis C virus; HIV, Human Immunodeficiency Virus; RA, Rheumatoid arthritis; CKD, Chronic kidney disease; TCAs, Tricyclic antidepressants; SSRIs, Selective serotonin receptor inhibitors; PPIs, proton pump inhibitors; *, *p* < 0.05; **, *p* < 0.01.

**Table 4 viruses-15-01021-t004:** Risk of osteoporosis in HPV-infected patients with or without treatments for HPV infections.

	Osteoporosis						
Variable	*n*	PY	IR	cHR	(95% CI)	aHR	(95% CI)
Non-HPV	277	302,947	0.91	1.00		1.00	
Treatment for HPV infections							
with treatment †	31	32,546	0.95	1.18	(0.81,1.71)	1.14	(0.78,1.66)
without treatment *	85	59,346	1.43	1.39	(1.09,1.77) **	1.40	(1.09,1.80) **

PY, person-year; IR, Incidence rate; cHR, Crude Hazard ratio; aHR, Adjust Hazard ratio. † “with treatment” group: HPV infected patients received any HPV-related treatment procedures (Cryotherapy, Electrocautery, Excision, Laser surgery) within three months after the index date. * “without treatment” group: HPV infected patients received no HPV-related treatment procedures or any procedures beyond three months after the index date; **, *p* < 0.01.

## Data Availability

Data were acquired from the NHIRD in Taiwan. The NHI database was available upon application.

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
