# Peer review of "Human Papillomavirus Infections and Increased Risk of Incident Osteoporosis: A Nationwide Population-Based Cohort Study"

_viruses, 2023, doi:10.3390/v15041021_

Round 1

Reviewer 1 Report

L94-95. Authors used data from Taiwan’s National Health Insurance Research Database (NHIRD) . Such databases usually contain diagnoses as ICD 10 codes and no results of lab tests. The reason is that physicians get payment from the insurance companies for the test but independent of test result (negative/positive) . Did authors used any ICD 10 code or something else for evaluation HOV infection?

L110-112  Only patients with the diagnosis of HPV from at least one inpatient admission or three outpatient visits were selected. The index date was defined as the date of HPV infections diagnosis

When I take the situation in Germany, HPV virus is mainly tested in outpatient offices like gynecologists (by women), using labor methods. Either physicians perform lab investigation in own practice or they send to lab and get back the results within few days. With other words, HPV diagnosis documented by an outpatient visit should be valid. Why did authors decide to take this diagnosis as valid if at least three outpatient visits were selected. This does not seem logically for me.  IS there a problem with the quality of outpatient diagnoses in Taiwan? This should be explained.

Table 2 and text in results. Retrospective studies do not allow the conclusions about risk factors, but only about associations. Risk factors need causal relationships which cannot be seen in sch studies. Correct term would be “Factors associated with a diagnosis of osteoporosis”. Please revise the whole manuscript to avoid the term ‘risk’.

Table 2 and methods: authors should explain in methods why exactly these covariables were included.

Author Response

We appreciate your comment and have taken it into consideration. We have revised our introduction and table 2 in accordance with your feedback. With regards to the question about diagnoses, we utilized ICD codes and CPT code to assess HPV infection, and the study population was selected based on previous research studies.

Chen, Huang-Hsi et al. “Risk of primary Sjogren's Syndrome following human papillomavirus infections: a nationwide population-based cohort study.” Frontiers in immunology vol. 13 967040. 16 Aug. 2022, doi:10.3389/fimmu.2022.967040

Juang, Sin-Ei et al. “Human Papillomavirus Infection and the Risk of Erectile Dysfunction: A Nationwide Population-Based Matched Cohort Study.” Journal of personalized medicine vol. 12,5 699. 27 Apr. 2022, doi:10.3390/jpm12050699

Reviewer 2 Report

·    The introduction should be improved regarding the mechanism through which viral infections, particularly those sustained by HPV, act their role in osteoporosis. 

·        Line 110: Should the Authors clarify the definition of “clinically diagnosed with HPV”? How were the men recruited and which type of lesions were considered? 

·        Should the Authors specify the mean follow-up period by cohort study?

·        It is unclear whether the authors considered HPV positivity alone or only cases with the presence of (pre)cancerous lesions. This is a relevant point considering that the immune response varies depending on if the infection is transient or if referred to a carcinogenesis process. A stratified analysis based on the degree of lesion would be more appropriate.

Author Response

We appreciate your comment and have taken it into consideration. We have revised our introduction and limitation in accordance with your feedback. With regards to the question about diagnoses, we utilized ICD codes and CPT code to assess HPV infection, and the study population was selected based on previous research studies. Unfortunately, we don't have the mean follow-up period data available.

Round 2

Reviewer 1 Report

N/A

Author Response

N/A

Reviewer 2 Report

I thank the Authors for their response. However, there are several points that the Authors did not answer appropriately.

- Despite the large sample-size, it remains unclear the definition of HPV-positive, especially which type of lesions were present.
- I invite the Authors to provide point-by-point responses
- in addition, the authors should limit the number of citations to 27 and 28 to describe the enrollment methods.

Author Response

Thank you for your feedback on our manuscript. We appreciate your constructive feedback, which has helped us improve the clarity and focus of our research.

-Despite the large sample-size, it remains unclear the definition of HPV-positive, especially which type of lesions were present.

Response: We apologize for any confusion caused by our unclear definition of HPV-positive in the initial submission. We acknowledge our mistake in not including the ICD code initially. In response to your comment, we have included the relevant ICD code (including the types of lesions) to clarify the definition of HPV-positive.

- in addition, the authors should limit the number of citations to 27 and 28 to describe the enrollment methods.

Response: In response to your suggestion, we have revised our manuscript to limit the number of citations in the methods section.